# Tumor cell intrinsic and extrinsic features predict prognosis in estrogen receptor positive breast cancer

**Kevin Yao**[1], **Evelien Schaafsma**[2,3], **Baoyi Zhang**[4], **Chao Cheng**[5,6,7] *

**1** Department of Electrical and Computer Engineering, Texas A&M University, College Station, Texas, United States of America, **2** Department of Molecular and Systems Biology, Dartmouth College, Lebanon, New Hampshire, United States of America, **3** Department of Biomedical Data Science, The Geisel School of Medicine at Dartmouth College, Lebanon, New Hampshire, United States of America, **4** Department of Chemical and Biomolecular Engineering, Rice University, Houston, Texas, United States of America, **5** Department of Medicine, Baylor College of Medicine, Houston, Texas, United States of America, **6** Dan L Duncan Comprehensive Cancer Center, Baylor College of Medicine, Houston, Texas, United States of America, **7** Institute for Clinical and Transcriptional Research, Baylor College of Medicine, Houston, Texas, United States of America

* chao.cheng@bcm.edu

**Data Availability Statement:** All datasets included in this study are publicly available at The Cancer Genome Atlas (TCGA) via FireHose (http://gdac.broadinstitute.org/), the European Genome Phenome Archive with accession ID

## Abstract

Although estrogen-receptor-positive (ER+) breast cancer is generally associated with favorable prognosis, clinical outcome varies substantially among patients. Genomic assays have been developed and applied to predict patient prognosis for personalized treatment. We hypothesize that the recurrence risk of ER+ breast cancer patients is determined by both genomic mutations intrinsic to tumor cells and extrinsic immunological features in the tumor microenvironment. Based on the Cancer Genome Atlas (TCGA) breast cancer data, we identified the 72 most common genomic aberrations (including gene mutations and indels) in ER+ breast cancer and defined sample-specific scores that systematically characterized the deregulated pathways intrinsic to tumor cells. To further consider tumor cell extrinsic features, we calculated immune infiltration scores for six major immune cell types. Many individual intrinsic features are predictive of patient prognosis in ER+ breast cancer, and some of them achieved comparable accuracy with the Oncotype DX assay. In addition, statistical learning models that integrated these features predicts the recurrence risk of patients with significantly better performance than the Oncotype DX assay (our optimized random forest model AUC = 0.841, Oncotype DX model AUC = 0.792, p = 0.04). As a proof-of-concept, our study indicates the great potential of genomic and immunological features in prognostic prediction for improving breast cancer precision medicine. The framework introduced in this work can be readily applied to other cancers.

## Author summary

Many genomic biomarker tests such as Oncotype DX have been developed for breast cancer and have helped guide clinical decisions. We have developed gene signatures to

EGAS00000000083, and the Gene Expression Omnibus (GEO) under accession numbers GSE41994, GSE101780, GSE3494, GSE22093, GSE22358, GSE22597, and GSE47561.

**Funding:** This work is supported by the Cancer Prevention Research Institute of Texas (CPRIT) (RR180061 to CC) and the National Cancer Institute of the National Institutes of Health (1R21CA227996 to CC). CC is a CPRIT Scholar in Cancer Research. The funders had no role in study design, data collection and analysis, decision to publish, or preparation of the manuscript.

**Competing interests:** The authors have declared that no competing interests exist.

integrate cancer genomic and transcriptomic data to characterize the downstream effect of driver genomic events. These signatures recapitulate the de-regulated pathways underlying the corresponding driver genomic events and are more correlated with clinical phenotypes such as recurrence free survival than mutation status alone. We apply this framework to ER+ breast cancer and define gene signatures for a total of 72 most commonly observed genomic events including gene mutations, amplifications and deletions. We find that many of these gene signatures are predictive of patient prognosis in ER+ breast cancer, and some of them achieved comparable accuracy with the Oncotype DX assay. We combine these tumor-intrinsic signatures with infiltration signatures for major immune cell types (tumor-extrinsic features) to construct integrative models for prognosis prediction. The models predicts the recurrence risk of patients with significantly better performance than the Oncotype DX assay.

## Background

Breast cancer is the leading cause of cancer in women worldwide. In 2021, it is estimated that 284,200 new patients will be diagnosed and 44,130 will die from breast cancer in the USA [1]. Breast cancer patients are often grouped by estrogen receptor (ER) and progesterone receptor (PR) status. Approximately 80% of breast cancers are estrogen receptor-positive (ER+) [2], and these patients show better response to endocrine therapy and have favorable prognosis as compared to ER-negative breast cancer [3]. Regardless, there exists a wide disparity in prognosis within ER+ breast cancer patients. A subtype of ER+ breast cancer patients with an insensitivity to endocrine therapy has been reported, involving complex interactions between the human epidermal growth factor receptor-2 (HER2), ER, and other signaling pathways [2,4]. In fact, the PAM50 gene expression test for intrinsic cancer subtyping has been shown to provide greater prognostic information than immunohistochemistry (IHC) for ER status or clinical variables [5]. For example, some ER+ patients with Luminal A breast cancer will live for over 10 years without experiencing breast cancer recurrence when treated with adjuvant tamoxifen despite exhibiting high grade, lymph node invasion, and overall higher recurrence risk, while other ER+ patients in the Basal subtype all relapse within 5 years [5]. This variance in prognosis even within the ER+ breast cancer subtype has motivated significant efforts to develop gene signatures to predict clinical outcomes and provide personalized treatment.

Indeed, a number of gene signatures have been developed to predict prognosis and stratify patients. The Oncotype DX Breast Recurrence Score has been the most widely validated gene signature in predicting prognosis for ER+ breast cancer patients, considering metrics such as its clinical utility and its impact on decision making [6]. The assay assigns a score from 0–100 (Onco-score) based on a breast cancer biopsy which classifies a patient as having a low (Onco-score: 0–17), intermediate (Onco-score: 18–30), or high (Onco-score: 31–100) risk for distal metastasis. The analysis is based on 21 genes with specific functions in tumor proliferation or invasion, HER2, and hormone receptor [7]. These 21 genes were selected based on their high association between gene expression levels and patient recurrence. Patients classified as high risk by the Oncotype DX Assay have been shown to experience a significantly lower risk for distal recurrence when treated with chemotherapy, whereas patients in the low or intermediate risk category experience little benefit from chemotherapy [8]. Interestingly, for patients treated with tamoxifen, a form of endocrine therapy, patients classified with a low or intermediate risk experience a significantly improved distal metastasis-free survival (DMFS) rate whereas patients in the high-risk category have a smaller benefit [9]. Another gene signature,

MammaPrint, has also been well-studied and has shown efficacy in predicting both ER+ and ER- breast cancer prognosis. It is based on 70 genes that were chosen using a "leave-one-out" strategy from 231 genes that were significantly associated with patient prognosis [10]. The MammaPrint signature classifies patients into good or poor signature groups based on their risk for distal metastasis. Similar to Oncotype DX, patients with high risk experience significant benefit from chemotherapy, whereas those in the low risk category do not [11]. It is evident that prognostic prediction and the discovery of risk groups via genomic assays can guide therapeutic decisions.

A number of cancer driver genes, including oncogenes and tumor suppressors, are frequently altered through somatic mutations and copy number variations (CNVs). Such genomic events are responsible for the initiation, progression and metastasis of tumors [12,13]. For example, the *TP53* oncogene has functions in tumor suppression and DNA repair and is the most commonly mutated gene in cancer: depending on cancer type, up to 50% of cancer cases have a somatic mutation in the *TP53* gene [14]. Specifically, 25% of breast cancer patients have somatic *TP53* mutations [14]. Mutations in driver genes leads to aberrant activation (oncogenes) or inactivation (tumor suppression), which in turn deregulate downstream oncogenic pathways. However, genomic aberrations such as somatic mutations and CNVs are often only weakly associated with prognosis [15–18]. As an example, the prognostic value of the *TP53* gene mutation is inconsistent and sometimes controversial in breast cancer [19,20]. Numerous different mechanisms can deactivate the p53 pathway besides *TP53* mutations, such as hypermethylation, CNV, or mutation of other genes in the p53 pathway [21], convoluting the impact of *TP53* mutations on oncogenic pathways. Gene signatures that recapitulate the downstream pathways of p53 mutations have been proposed as better prognostic markers [22–25]. Signatures for other genes have also been developed to predict prognosis, such as *PIK3CA* [26,27], *BRAF* [26], *KRAS* [26], *TMPRSS-ERG* [28], etc. However, the prognostic value of signatures for all common genomic alterations have not been investigated in a systematic manner.

Genomic aberration of driver genes occurring in cancer cells represent a set of tumor-intrinsic features. In addition to genomic changes intrinsic to cancerous cells, patient prognosis is also determined by immunological features in the tumor microenvironment (TME). Tumor-infiltrating immune cells interact with the TME through a complex process known as cancer immunoediting, which involves both immunologic clearance of cancer cells and the promotion of non-immunogenic cancer clones that edits the overall immunogenicity of the tumor [29]. Somatic mutations in cancer cells can be presented as neoantigens that can be recognized by T cells to trigger an immune response. Thus, an increase of CD8+ and CD4+ T-cell infiltration in the TME is correlated with better prognosis across cancer types, including colorectal [30], lung [31], and breast cancers [32]. However, the presence of regulatory T cells serves to facilitate tumor escape and expansion, worsening prognosis [33]. Although less commonly studied, infiltrating B cells in the tumor microenvironment have also been correlated with good prognosis in melanoma [34] and multiple subtypes of breast cancers [35]. Natural Killer cells, despite functioning as tumor cell killers, have also been correlated with advanced disease and may facilitate cancer growth and poor prognosis [36,37]. Lastly, increased macrophagic infiltration has been correlated with favorable clinical outcomes [38]. These various effects of immune infiltration on prognosis highlight the importance of such data in prognosis prediction models.

In this study, we aim to systematically investigate the prognostic value of tumor-intrinsic and -extrinsic features in ER+ breast cancer. Particularly, we define gene signatures for 72 genomic aberrations, including somatic mutations in *TP53*, *PIK3CA*, *CDH1*, and *GATA3*, which comprise tumor-intrinsic features. We also infer the immune infiltration of 6 immune cells, which comprises tumor-extrinsic features. We then construct predictive models that

integrate these genomic features (Sig model), immunological features (Imm model), and a combination of both (Sig+Imm model). We compare the predictive power of these models with Oncotype DX scores as a reference with or without incorporating established clinical variables (e.g. age, tumor stage, etc.). Our results indicate that many of our individual signatures and immune cell infiltration scores are prognostic when used as solo predictors. We also find that an optimized model that integrates both intrinsic and extrinsic features achieves a significantly higher prediction accuracy than Oncotype DX. Our study provides a generic framework to define integrative models based on genomic data to combine intrinsic and extrinsic features for predicting clinical outcomes.

## Results

### Overview of this study

Cancer is a genetic disease, in which mutations of oncogenes or tumor suppressors lead to the aberrant activation or inactivation of specific oncogenic pathways and eventually deregulated gene expression (see examples shown in Fig 1A). Previous studies have shown that gene signatures modeling deregulated path ways better predict prognosis than the corresponding gene mutations. In this study, we develop a statistical framework to systematically investigate a comprehensive list of genomic events of genes frequently observed in ER+ breast cancer, including 4 mutations, 53 amplifications, and 15 deletions (Table 1). By using gene-specific multivariate regressions, we modeled the combinatorial effect of these genomic events in regulating gene expression in TCGA ER+ breast cancer samples. Based on these models, we defined a gene signature for each of these genomic events. These gene signatures can be used to calculate sample-specific scores that indicate downstream pathways associated with the corresponding genomic events. We rationalized that these signatures will provide a systematic characterization of deregulated pathways intrinsic to tumor cells and are thus informative for predicting patient prognosis in ER+ breast cancer.

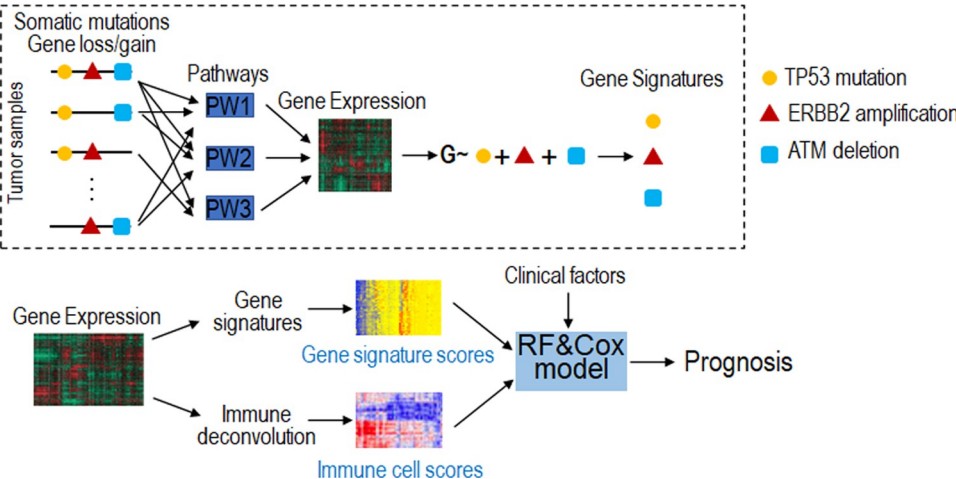

**Fig 1. Schematic diagram of this study. a** Definition of gene signatures to recapitulate the pathways underlying driver genomic aberrations. Here we use three genomic aberrations (*TP53* mutation, *ERBB2* amplification and *ATM* deletion) as examples. In ER+ breast cancer, we defined a total of 72 gene signatures, each for a specific genomic aberration. **b** To predict patient prognosis in ER+ breast cancer, we constructed prediction models to integrate the 72 gene signatures (intrinsic features), 6 types of infiltrating immune cells (extrinsic features), and clinical factors (e.g., age, tumor stage). Gene signature scores and immune cell scores were calculated based on gene expression of tumor samples. Random Forest models were used to classify good versus poor prognosis, and Cox regression models were used to predict prognostic risk scores.

**Table 1. Summary of features in each model discussed in the manuscript.**

| Feature Type | Features |
|---|---|
| *Clinical* | Age, Stage, Lymph node status, Grade, Size |
| *Signature (somatic mutation)* | *CDH1, GATA3, PIK3CA, TP53* |
| *Signature (gene gain)* | *ABL2, ARNT, BRIP1, CCND1, CD79B, CDK12, CIITA, CLTC, COL1A1, COX6C, CREBBP, DDX5, ELK4, ERBB2, ERCC4, EXT1, FCGR2B, FGFR1, FH, FUS, GNAS, H3F3A, HEY1, HLF, HOOK3, IL21R, MDM4, MSI2, MUC1, MYC, MYH11, NCOA2, NDRG1, NTRK1, PALB2, PBX1, PLAG1, PRKAR1A, PTPRC, RAD21, RNF43, SDC4, SDHC, SOCS1, SPOP, SRSF2, SS18L1, TCEA1, TNFRSF17, TPM3, TPR, TSC2, WHSC1L1* |
| *Signature (gene loss)* | *ARHGEF12, ATM, BIRC3, CBFA2T3, CDH1, CYLD, FLI1, HERPUD1, MAF, MAP2K4, PCM1, PCSK7, POU2AF1, SDHD, WRN* |
| *Immunological* | Naïve B (NavB), Memory B (MemB), CD4+ T (CD4T), CD8+ T (CD8T), natural killer cells (NKcell), and Monocytes |
| *Random Forest optimized model* | Size, lymph_nodes_positive, age_at_diagnosis, TP53_mut, ABL2_amp, CCND1_amp, CD79B_amp, CIITA_amp, CLTC_amp, COX6C_amp, ELK4_amp, FH_amp, MSI2_amp, MYC_amp, NCOA2_amp, NTRK1_amp, PLAG1_amp, SDC4_amp, SS18L1_amp, TCEA1_amp, TNFRSF17_amp, MemB |
| *Cox optimized model* | Size, lymph nodes status, age at diagnosis, ELK4_amp, CCND1_amp, NTRK1_amp, CREBBP_amp, MAP2K4_del, PCM1_del, SDHD_del, PCSK7_del, MemB, SOCS1_amp, DDX5_amp, SDHC_amp, MSI2_amp |

To further include tumor-extrinsic features, we considered the infiltration levels of six major types of immune cells in the tumor microenvironment, which can also be calculated based on gene expression profiles of tumor samples. We then developed prediction models to integrate these intrinsic and extrinsic features as well as established clinical factors to predict prognosis in ER+ breast cancer (Fig 1B). Specifically, classification models based on Random Forest were constructed to classify patients with good versus poor prognosis; and Cox proportional hazards regression were constructed to predict prognostic risk of patients and support the Random Forest results. Using this framework, we can systematically characterize potential tumor intrinsic and extrinsic features fully based on transcriptomic data for prognostic prediction.

## Gene signature scores recapitulate the mutation status of driver genes

Based on the TCGA data, we defined a total of 72 gene signatures to characterize all frequently occurring genomic aberrations in ER+ breast cancer. Patients with higher gene signature scores have expression profiles with a greater degree of similarity to the expression profile of patients with the genomic aberration. To investigate the interdependence between different gene signatures, we calculated pairwise Spearman correlation coefficients (SCCs). Most of these signatures were weakly correlated (Fig 2A), indicating that they capture different downstream signaling outputs.

To examine whether these gene signatures recapitulate their genomic events, we applied them to independent ER+ breast cancer expression datasets that provided gene mutation status. As shown in Fig 2B, the *TP53* mutation (TP53_mut) signature scores were significantly higher in *TP53* mutant samples than wild-type samples (p = 1e-12). Similarly, for another three genomic aberrations (*HER2* amplification, *PIK3CA* mutation, and *GATA3* mutation), patients with the aberration displayed significantly higher signature scores than wild-type patients (Fig 2C, 2D and 2E). To quantify the accuracy of signature scores in classifying mutant versus wild-type samples, we determined the receiver operating characteristic (ROC) curves for the TP53_mut (Fig 2F) and ERBB2_amp (Fig 2G) signatures in multiple ER+ breast cancer

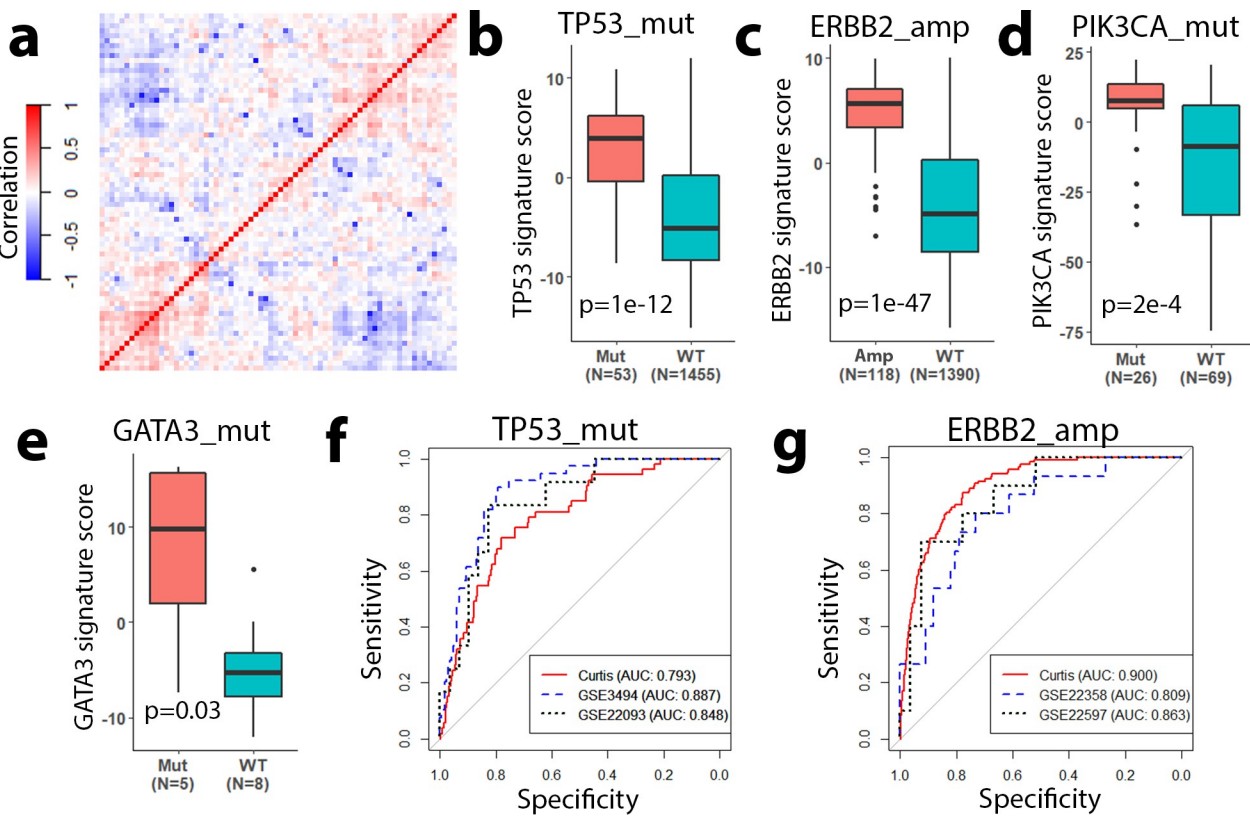

**Fig 2. Gene signatures recapitulate the downstream pathways of mutated driver genes. a** Spearman correlation coefficients (Correlation) between different gene signatures defined based on the TCGA ER+ breast cancer data. Signature scores can distinguish ER+ breast cancer samples with TP53_mut (**b**), ERBB2_amp (**c**), PIK3CA_mut (**d**), and GATA3_mut (**e**) from samples without the aberrations. **b, c** were based on the Curtis data; (**d**) was based on GSE41994; and (**e**) was based on GSE101780. ROC curves showing that TP53_mut (**f**) and ERBB2_amp (**g**) signature scores can predict the mutation status of their respective driver genomic aberration.

datasets (*TP53*: GSE3494 and GSE22093, *ERBB2*: GSE 22358 and GSE22597). High area under the receiver operating characteristic curve (AUC) scores are observed (AUC>0.8), indicating that our gene signature scores are able to recapitulate information on the aberration status of driver genes in validation datasets.

Since these signatures were defined based on ER+ breast cancer data, we expected that they might only be effective in ER+ breast cancer. Indeed, when applied to ER- breast cancer, these signatures poorly discriminated mutant versus wild-type samples as exemplified by the *TP53* and *ERBB2* signatures (S1 Fig).

## Association of gene signatures and immune cell infiltration with patient prognosis

We then analyzed the distribution of individual signature scores based on patient prognosis, reasoning that individual signature scores must be differently distributed to be able to predict prognosis. In the Curtis dataset, which contains microarray data for 1508 ER+ breast cancer patients [39], we stratified ER+ breast cancer patients into a good (alive after 10 years) and poor (death due to disease before 10 years) prognosis group with 387 and 293 patients, respectively. For each of the 72 gene signatures, we examined the association with patient prognosis and found that 52 gene signatures were prognostic (S1 Table). For example, TP53_mut signature scores were significantly higher in patients with poor prognosis as compared to those with

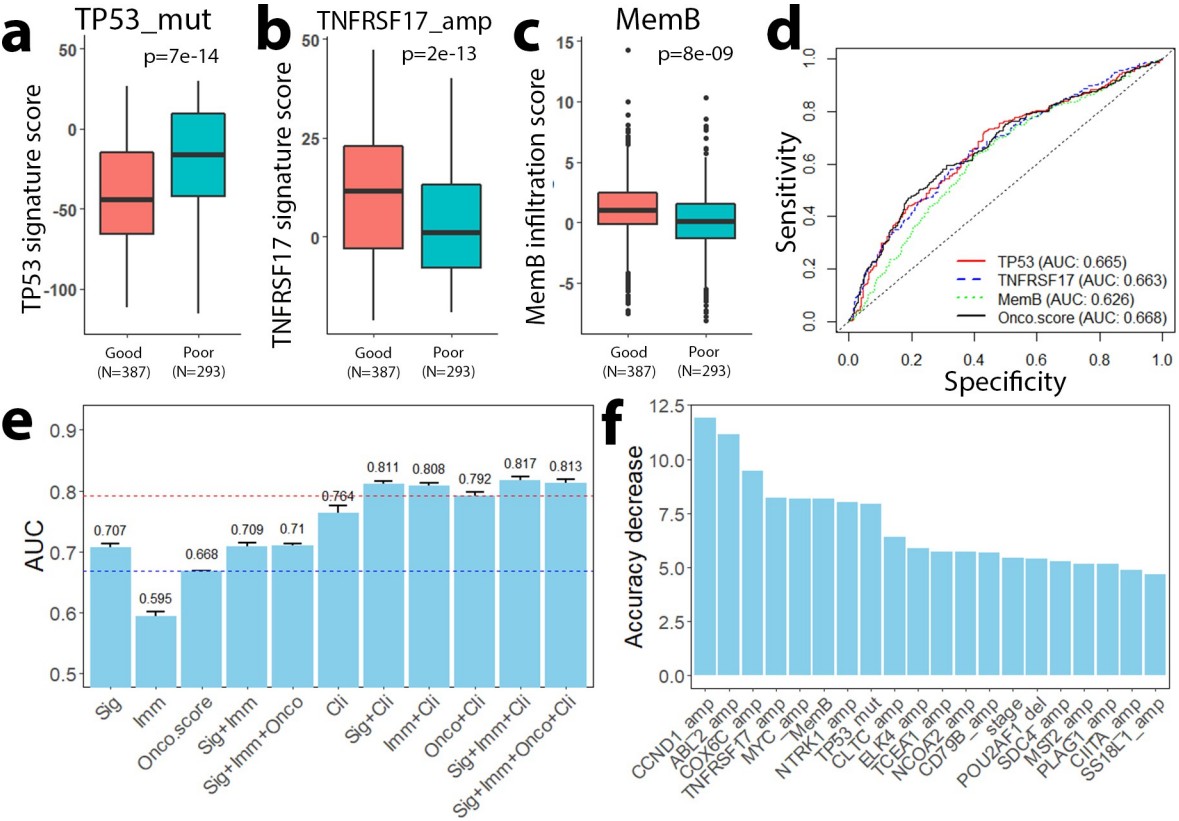

**Fig 3. Gene signatures and immune infiltration scores predict patient prognosis. a-c** Signature scores for TP53_mut (**a**), TNFRSF17_amp (**b**), and MemB (**c**) distinguish patients with good and poor prognosis. **d** ROC curves showing that TP53_mut, TNFRSF17_amp and MemB infiltration score predicts prognosis at a comparable level to Onco-score. **e** AUC scores of random forest models with different combinations of predictive features. Our Sig+Imm model performs with higher AUC scores than the Onco-score models with and without clinical features. **f** Relative importance of the top 20 most important genomic aberration and immune infiltration features.

good prognosis (p = 7e-14, Fig 3A). This is consistent with previous reports that mutations in the *TP53* gene and disruption of the p53 pathway were correlated with poorer prognosis [40–42]. Similarly, the TNFRSF17_amp signature scores were significantly different between groups with higher values in patients with good prognosis (p = 2e-13, Fig 3B). Such an association has not been previously reported in ER+ breast cancer, but its prognostic value has been studied for ER- breast cancer [43–45]. *TNFRSF17* is an immune response gene with overexpression correlating with better prognosis [43].

In additional to these gene signatures, we also investigated the prognostic value of infiltration levels of six types of immune cells, including Naïve B (NavB), Memory B (MemB), CD4 + T (CD4T), CD8+ T (CD8T), natural killer cells (NKcell), and Monocytes (see S2 Table). Based on the inferred infiltration scores, we observed a correlation between MemB cell infiltration and patient prognosis in ER+ breast cancer: patients with good prognosis had significantly higher MemB cell infiltration scores than those with poor prognosis (p = 8e-9, Fig 3C).

For all gene signatures and infiltrating immune cell types, we calculated AUC scores to quantify the ability to classify good versus poor prognosis groups in ER+ breast cancer (S1 and S2 Tables). Specifically, the AUC scores for TP53_mut, TNFRSF17_amp and MemB cells were 0.665, 0.663 and 0.626, respectively, which was comparable to the accuracy achieved by the widely used Onco-score (AUC = 0.668) (Fig 3D). All together, our results indicated that both

intrinsic gene signatures and extrinsic infiltrating immune cells were informative for predicting patient prognosis in ER+ breast cancer.

## Integrative models for classifying good and poor prognosis patient groups

After showing the clinical significance and prognostic power of our individual signature scores, we constructed RF models with different combinations of scores to classify patient prognosis. We then determined their performance by 10-fold cross-validation in the Curtis data (Fig 3E). First, we constructed three RF models that integrated the 72 gene signatures (Sig), the six immune cell types (Imm) and a combination of both features (Sig+Imm). The Sig model achieved an accuracy of AUC = 0.707, which was higher than the accuracy achieved by the Onco-score (AUC = 0.668). In contrast, the Imm model had a relatively low AUC (AUC = 0.595). In addition, the Sig+Imm model had similar accuracy as the Sig model, suggesting that adding immunological features does not further increase the performance of the Sig model. In fact, breast cancer has been reported as immune cold and the overall response rate of ER+ breast cancer patients to immunotherapy is about 12% [46,47]. Second, we further incorporated several clinical factors (age, tumor size, grade, stage and lymph node status) to construct the Sig+Clin, Imm+Clin, Sig+Imm+Clin models. The AUC scores of these models (0.811, 0.808 and 0.817, respectively) were higher than the accuracy of the Clin model (AUC = 0.764), which was solely based on clinical factors. Moreover, the accuracies of these models were also higher than the Onco+Clin model (AUC = 0.792), suggesting that both genomic features and immunological features provide additional prognostic value that surpass the predictive ability of the Oncotype DX assay. Of note, incorporating the Onco-score (the Sig +Imm+Onco+Clin model) did not further improve the classification accuracy, suggesting that information provided by the Onco-score is captured by the genomic and/or immunological features. Thus, our gene signatures and immune infiltration scores may encompass information contained in the Oncotype DX assay and provide additional prognostic prediction potential.

To identify features that contributed most to the prediction ability, we examined the relative importance of all features included in the Sig+Imm+Clin RF model. As shown in Fig 3F, the top 10 most important Sig or Imm features are CCND1_amp, ABL2_amp, COX6C_amp, TNFRSF17_amp, MYC_amp, MemB, NTRK1_amp, TP53_mut, CLTC_amp, and ELK4_amp. A complete list of the relative importance for all features can be found in S3 Table. Out of the most predictive features, many have been studied previously as prognostic factors in breast cancer, such as CCND1_amp [48], ABL2_amp [49], TP53_mut [50], TNFRSF17_amp [43–45], and memory B infiltration [51]. However, there are other important gene aberrations like COX6C_amp that have not been previously reported as prognostic. Our models thus shed light on the roles of various unexplored mutations/CNVs in breast cancer prognosis.

## An optimized model outperforms Oncotype DX scores for prognosis classification

We optimized the Sig+Imm+Clin model by iteratively removing the least important features from the model and then recalculating the relative importance of the remaining features (Fig 4A). Eventually, we obtained an optimized model with a total of 22 (including 18 Sig, 1 Imm, and 3 Clin features) predictive features (Table 1). The accuracy of this optimized model was AUC = 0.841 according to cross-validation results in the Curtis data, which was much higher the Onco+Clin model (AUC = 0.791, p = 0.04) and the Clin model (AUC = 0.763, p = 2e-4) (Fig 4B). The significance test was performed using Delong's test for two correlated ROC curves. In addition, when trained in the Curtis discovery dataset and evaluated in the Curtis

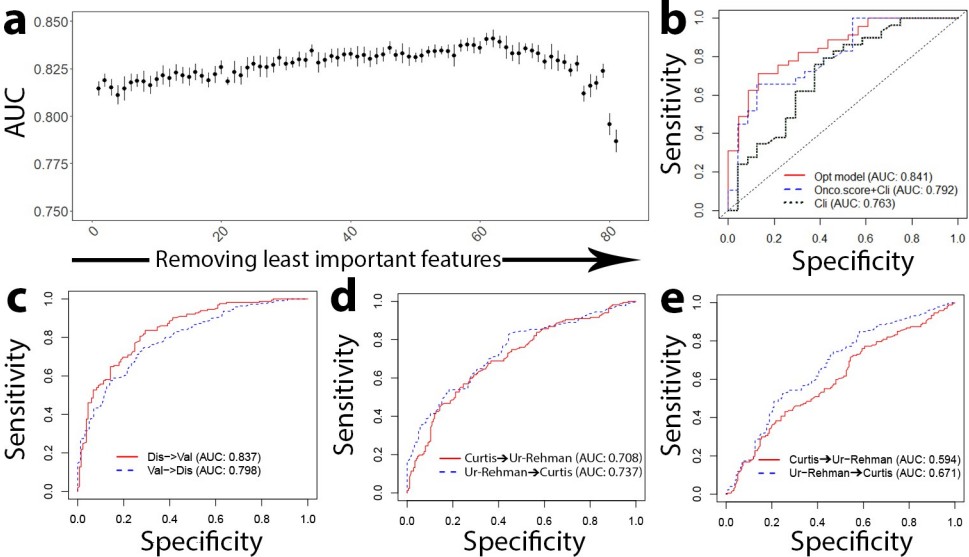

**Fig 4. Optimized model outperforms Oncotype DX risk scores for prognostic prediction. a** Results from backward selection to find an optimized set of features—AUC score of the model plotted as a function of the number of features removed. The optimized model is chosen as the highest AUC score at the smallest number of features. **b** ROC curves of the performance of our optimized model as compared to the Onco-score + Cli model and just the Cli model, showing that our optimized model overperforms both Oncotype DX and clinical features. **c** ROC curves of our optimized model when trained in Curtis discovery and validated in Curtis validation, and vise versa. **d** ROC curves of our optimized model when trained in Curtis discovery and validated in the test dataset, the Ur-Rehman dataset, and vise versa. **e** ROC curves of the Onco+Cli model trained and validated in the same way as (**d**), showing decreased performance compared to our optimized model.

validation data, the optimized model achieved an AUC of 0.837. Conversely, an AUC 0f 0.798 was obtained when the training and test datasets were swapped (Fig 4C).

To demonstrate the model's ability to predict prognosis in clinical situations, the optimized model was trained in a training dataset and then applied to an independent test dataset, the Ur-Rehman dataset, which contains curated data from 856 ER+ breast cancer patients [52]. When the optimized model was trained in the Curtis data and tested in the Ur-Rehman data, we observed an accuracy of AUC = 0.708, and vice versa, an accuracy of AUC = 0.737 (Fig 4D). This performance was significantly higher than that of the Onco+Clin model, which achieved AUC = 0.594 and AUC = 0.671 for Curtis-to-Rehman and Rehman-to-Curtis predictions, respectively (Fig 4E). Altogether, our results indicated that the optimized model achieved consistently higher performance than the Oncotype DX assay.

## Integrative models for predicting prognostic risk based on Cox regression

We constructed Cox proportional hazards models to further validate the prognostic values of the gene signatures and infiltrating immune cells. We applied univariate and multivariate Cox regression models to investigate the association between these individual features and patient disease-free survival with or without including clinical variables in the Curtis dataset. Using the gene signature scores or the immune infiltration scores as continuous variables, we found 50 features (46 gene signatures and 4 immune cells) that were significantly associated with patient survival without considering clinical variables (Fig 5A and S4 Table). After adjusting for clinical variables, 36 features (34 gene signatures and 2 immune cells) were significantly associated with patient survival (Fig 5B and S4 Table). These results indicated that many of gene signatures alone can stratify ER+ breast cancer patients into subgroups with different

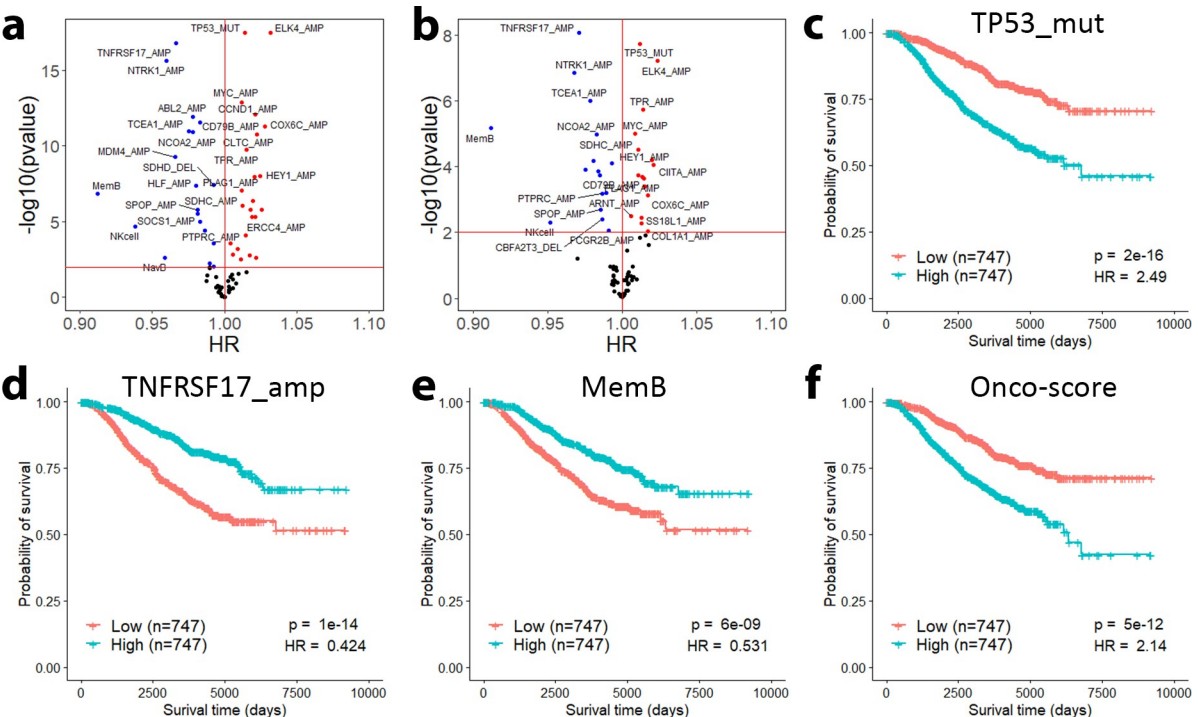

**Fig 5. Individual signature and immune infiltration scores can identify prognostic patient groups.** Signature and immune infiltration scores fitted in a univariate Cox proportional hazards model without clinical adjustment (**a**) and with clinical adjustment (**b**) are associated with survival time. Patients are significantly dichotomized by their TP53_mut (**c**), TNFRSF17_amp (**d**), and MemB (**e**) score. **f** Risk groups dichotomized by the median Onco-score have lower or comparable significance to some of our signature and immune infiltration scores.

prognosis. For example, patients with high TP53_mut score showed significantly shorter survival time than those with low score (p = 2e-16, Fig 5C). In contrast, patients with high TNFRSF17_amp (Fig 5D) or MemB infiltration scores (Fig 5E) had significantly prolonged survival. For comparison, we showed the survival curves of patients dichotomized based on the Onco-score (p = 5e-12, Fig 5F), which exhibited a lesser or comparable significance as compared to our signatures. Our results indicated that many gene signatures could give rise to better or comparable prognostic stratification than Onco-score in ER+ breast cancer.

Following this, we integrated all 78 signature scores and immune infiltration scores using multivariate Cox regression models and performed feature selection to obtain an optimized model with 16 variables. The selected features are listed in Table 1. We applied this optimized model to predict patient prognostic risk, which achieved a fairly high performance with a concordance index (CI) of 0.758 based on cross-validation in the Curtis data. Of note, a similar model based on clinical factors achieved a CI of 0.701, and the CI increased to 0.718 if Onco-type scores are further incorporated with clinical variables.

## An optimized Cox regression model for prognostic risk prediction

To demonstrate the clinical utility of the Cox-optimized model, we trained the model in the Curtis discovery dataset and then applied it to predict patient risk score in the Curtis validation dataset. Based on the predicted risk scores, we stratified patients into high-, intermediate- and low-risk groups of equal size. As shown in Fig 6A, the optimized model was overall able to separate patients into different risk groups (p = 1e-23). In particular, patients in the high risk category had a hazard 8.015 times that of the low risk category (p = 6e-20), and patients in the

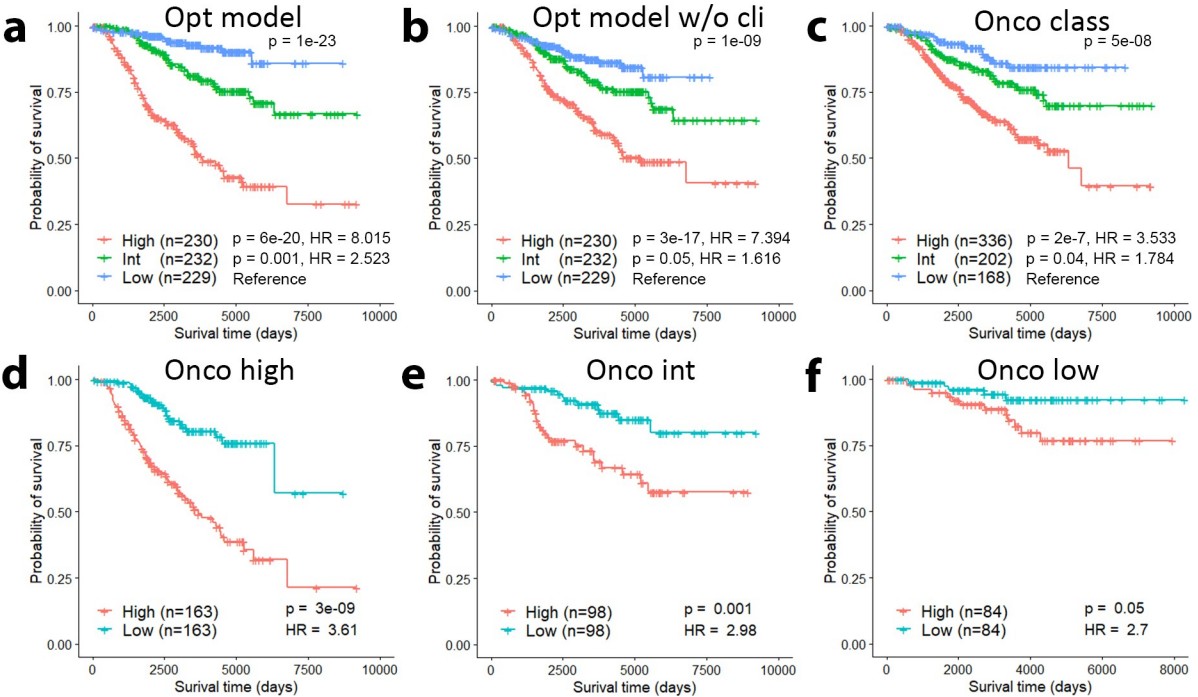

**Fig 6. Optimized Cox regression model for prognostic risk prediction. a** Patients in the Curtis validation dataset are significantly grouped by their risk as predicted by the optimized model trained in the Curtis discovery dataset. **b** Patients are still significantly dichotomized by their risk when clinical variables are removed from the optimized model. **c** Onco class achieves slightly lower performance than our optimized model without clinical information in grouping patient risk categories. **d-f** Our optimized model is able to further stratify the Onco high (**d**), intermediate (**e**), and low (**f**) risk classes.

intermediate risk category had a hazard 2.523 times that of the low risk category (p = 0.001). We also considered the predictive power of model after adjusting for the contributions of the clinical features by removing clinical features from the model (Fig 6B). The overall significance of the risk groups decreased moderately (p = 1e-09). Patients in the high risk group had a hazard 7.394 times that of the low risk category (p = 3e-17), and patients in the intermediate risk category had a hazard 1.616 times that of the low risk category (p = 0.05). The performance of our models in predicting prognostic risk were then compared to that of the three Onco-classes for patients in the Curtis validation dataset (Fig 6C). Overall, the Onco-classes slightly underperformed the optimized model adjusted for clinical features (p = 5e-8). Patients classified as high risk had a hazard 3.533 times that of patients in the low risk category (p = 2e-7), and those classified as intermediate risk had a hazard 1.784 times that of the low risk category (p = 0.04). Although the ability to distinguish low risk from intermediate risk patients was similar between Onco-class and our clinical variable-adjusted optimized model, our model was able to more significantly define a higher risk group from the intermediate risk group (p = 3e-5 for our model vs p = 3e-4 for Onco-class).

To investigate whether the optimized model has the potential to improve the Oncotype DX assay, we used our optimized model's risk predictions to further dichotomize patients in each Oncotype DX class. In each of the Onco high (Fig 6D), Onco intermediate (Fig 6E), and Onco low (Fig 6F) classes, our optimized model's risk prediction was able to separate patients into two statistically significant risk groups (Onco high: p = 3e-9, Onco int: p = 0.001, and Onco low: p = 0.05). To identify the prognostic power of our signature and immune scores alone, we repeated this test with the clinical variables removed (S2 Fig). We found that the Onco high

and Onco intermediate risk classes were significantly separated (p = 0.009 and p = 0.004, respectively) and the Onco low risk class was moderately separated (p = 0.2). These results show the potential of our models to be integrated in conjunction with Oncotype DX to provide more precise risk predictions.

## Discussion

In this study, we defined gene signatures for a comprehensive list of gene aberrations frequently observed in ER+ breast cancer to recapitulate their downstream regulatory pathways. These signatures were then used to calculate sample-specific scores based on tumor gene expression profiles. These scores represented a collection of tumor-intrinsic features that project gene expression to cancer-related pathway activities. The prognostic values of these signatures were validated by both Random Forest classification and Cox regression models. The performance of these prediction models that integrated gene signatures and immune cell infiltration scores were evaluated with or without including clinical variables. Our results indicated that these features have the great potential to further improve the prediction accuracy achieved by the Oncotype DX assay.

A number of genomic aberrations, such as *TP53* mutation and *ERBB2* amplification, have been frequently observed in breast cancer [53]. These genomic mutations lead to the deregulation of specific downstream pathways and confer selection advantage to tumor cells at certain stage of cancer development and progression. Essentially, it is the downstream pathways that drive tumorigenesis and determine clinical outcomes. The frequently observed gene mutations represent the most likely but not the only mechanism that deregulate the corresponding pathways. For example, it has been shown the p53 pathway can be inactivated not only by *TP53* mutation but also by alternative mechanisms like hypermethylation or mutation of other genes in the p53 pathway [21]. Therefore, the gene signatures defined by the proposed method provide a collection of candidate features that are prognostic. Importantly, a tumor sample may harbor multiple driver genomic mutations. Some of the driver genomic mutations are correlated, presenting together or mutually exclusive. As such, we defined gene signatures for 72 frequent gene mutation events in a systematic manner, where all signatures are calculated together in a multivariate linear model. The resulting gene signatures takes into the correlations between different genomic events and the cross-talking in their downstream pathways, and therefore are expected to better predict prognosis when combined using integrative models. We have also defined gene signatures for the same set of genomic aberrations but in a separate fashion, where the gene signatures are defined based on only their value in a univariate linear model (see Methods). Such a definition most directly correlates with the aberration status of the driver gene, but the lack of consideration for the interdependence between signatures makes them unfit for multivariate prediction models. Indeed, integrative prediction models based on these signatures resulted in worse prediction accuracy compared with those simultaneously defined signatures.

In addition to the gene signatures designed for charactering tumor-intrinsic features, we also include immune infiltration scores to capture tumor-extrinsic features. However, our results indicated that the prognostic values contributed by these immunological features, with the exception of MemB, were relatively low compared with gene signatures and clinical factors. This might be explained by the fact that breast cancer is relatively immune cold compared to other cancers. Immune cytolytic activity, mutation burden, and neoepitopes load are often correlated with immune response, yet breast cancer patients generally have relatively moderate levels of these factors [17,54]. Indeed, the overall response rate of ER+, HER2 negative breast cancer to pembrolizumab, an antibody targeting programmed cell death-1/programmed death

ligand-1, is only about 12% [46]. Specific infiltrating immune cell subsets associated with patient prognosis are also low in ER positive breast cancer. Such immune cells include CD8+ T and CD4+ T, which have been extensively reported as good prognostic markers [55,56], whereas regulatory T cells are known to be poor prognostic markers [57]. While triple negative breast cancer most commonly exhibit high infiltrations of these tumor-infiltrating lymphocytes (TILs), ER+ patients generally have lower TIL infiltration levels [58], making infiltration levels difficult to be accurately inferred by immune deconvolution methods from gene expression data. In fact, TIL levels have been found to not correlate with overall survival rate for ER+ breast cancer patients [59]. Even so, the memory B immune infiltration score was included in the random forest optimized model. Including immune features may allow for better and more accurate feature optimization of the predictive model. In addition, they also provide critical insights about different immunological features in terms of prognostic prediction. Future work may investigate the prognostic potential of these tumor extrinsic immune cell infiltration scores in prognostic prediction models for immune hot cancers like melanoma, lung cancer, and acute lymphoblastic leukemia.

Our analysis showed that the clinical factors alone can achieve relatively high accuracy in prognostic prediction. The prognosis of ER+ breast cancer is largely determined by proliferation rate of tumors [60], which can be captured at least partially by the clinical variables, tumor size and stage. Nevertheless, the prognostic prediction accuracy can be further improved when the gene signatures and immune infiltration scores are used (the Sig+Clin and the Imm+Clin models). In particular, the Imm+Clin model showed an improved prediction accuracy than the clinical model, even though immunological features alone had fairly poor performance. Overall, these results indicate that additional biomarkers developed from genomic, molecular or immunological characterization of tumors can further improve prognostic prediction in ER+ breast cancer.

## Conclusions

In conclusion, we have proposed a framework to systematically extract both tumor-intrinsic and extrinsic features from gene expression data for integrative prediction of prognosis in ER + breast cancer. Using this framework, we assessed the prognostic values contributed by different categories of features as well as by different genomic aberration events. This framework can be readily applied to all cancer types for improving precision medicine.

## Methods

### Datasets used in this study

Level 3 processed RNA sequencing (RNA-seq) data for 1097 breast cancer patients was downloaded from The Cancer Genome Atlas (TCGA) via FireHose (http://gdac.broadinstitute.org/ ). The data was normalized in the RNA-seq by Expectation-Maximization (RSEM) format. The processed somatic mutation data and copy number variation (CNV) segments were downloaded as Mutation Annotation Format (MAF) and segmented copy number alterations (sCNA) files, respectively, from TCGA using FireHose.

Gene expression profiles of other datasets were obtained as follows. The first dataset is from Curtis et al. (METABRIC)[61] and was downloaded from the European Genome Phenome Archive with accession ID EGAS00000000083. This dataset consists of 1992 breast cancer patients in two cohorts, the discovery cohort (997 patients), and the validation cohort (995 patients). 1508 patients were ER+. The data was measured using the Illumina HT-12 v3 platform (Illumina_Human_WG-v3). This dataset included the mutation status of the *TP53* gene, determined by examining exons 2–11 for mutations and scoring them using Mutation

Surveyor software. It also included *HER2* amplification status (whether the patient was experiencing *HER2* gain or *HER2* neutral), which was determined based on their empirical expression distributions using MCLUST. Additional seven datasets were downloaded from the Gene Expression Omnibus (GEO) under accession numbers GSE41994, GSE101780, GSE3494, GSE22093, GSE22358, GSE22597, and GSE47561. GSE41994 contains *PIK3CA* mutation status and gene expression data from 95 ER+ breast cancer patients [62]. GSE101780 contains *GATA3* mutation status and gene expression data from 13 ER+ breast cancer samples [63]. GSE3494 [23] and GSE22093 [64] contain information on 251 and 103 distinct breast cancer samples, respectively, and also contain *TP53* mutation status. GSE22358 [65] and GSE22597 [66] contain data on *HER2* amplification status of 158 and 82 breast cancer samples, respectively. The Ur-Rehman dataset, GSE47561, combines 10 other breast cancer datasets and contains a total of 1570 samples, 856 of which are ER+ [67]. The expression was measured by the Affymetrix microarray platform.

## Identification of frequently mutated or amplified/deleted genes in ER + breast cancer

MAF somatic mutation data from TCGA contained the number of nonsynonymous mutations in each gene. We considered all genes with mutations in which there was at least one nonsynonymous point mutation in at least 10 percent of all ER+ samples.

CNV data from TCGA contained genomic segments that significantly deviated from diploid (as in normal tissues) and their copy numbers in each tumor samples. Based on these data, copy numbers of genes were calculated by referring to the Ensemble human genome annotation file, which contains the genomic localization of genes. We $\log_2$ transformed these numbers to obtain the fold change of each gene's copy number. Genes that showed amplifications with a fold increase greater than $\log_2\left(\frac{2.8}{2}\right)$ or deletions with a fold decrease less than $\log_2\left(\frac{1.3}{2}\right)$ were selected for further investigation. A list of genes commonly exhibiting mutations and CNVs based on our approach are shown in Table 1, along with the type of genomic aberration occurring in that gene.

## Definition of weighted gene signatures for genomic events

For each recurrent genomic event (gene mutations, amplifications and deletions), we defined a weighted gene signature based on TCGA ER+ breast cancer RNA-seq data. The expression levels of genes (originally represented as RSEM) were adjusted based on the formula $\log_2$(RSEM+1) to avoid extreme values. First, the status of each genomic aberration $j$ in Table 1 ($X_j$ = 1 if there is an aberration, 0 if no aberration) is used in a multivariate linear model (Eq 1) as predictive variables, and the log2(RSEM+1) expression of gene $i$ in the gene expression profile is the response variable ($Y_i$) (Eq 1).

$$Y_i = \beta_{i,0} + \beta_{i,1}X_1 + \cdots + \beta_{i,j}X_j + \cdots + \beta_{i,72}X_{72} \qquad \text{Eq1}$$

For each genomic aberration $j$ in Table 1, the linear coefficients ($\beta_{i,j}$) and p value ($p_{i,j}$) for each gene $i$ in the gene expression profile is calculated. Two sets of weights are then calculated for each combination of genomic aberration $j$ and gene $i$ in the gene expression profile, $w_{i,j}^+$ and $w_{i,j}^-$. If the expression of gene $i$ in the gene expression profile is positively correlated with the presence of genomic event $j$ ($\beta_{i,j}>0$), then $w_{i,j}^+ = -\log p_{i,j}$ and $w_{i,j}^- = 0$. For negative correlations ($\beta_{i,j}<0$), then $w_{i,j}^- = -\log p_{i,j}$ and $w_{i,j}^+ = 0$. The weights are then normalized by capping the maximum weight at 10 and dividing by the range to transform the weights to a decimal between zero and one. The result of this definition of $w_{i,j}^+$ and $w_{i,j}^-$ is the following: comparing

patients with and without genomic aberration $j$, if gene $m$ is more upregulated than gene $n$, then $w_{m,j}^+ > w_{n,j}^+$ and $w_{m,j}^- = 0$. Similarly, if gene $m$ is more downregulated than gene $n$, then $w_{m,j}^- < w_{n,j}^-$ and $w_{m,j}^+ = 0$.

The above gene signatures take co-occurrences and mutual exclusiveness among different genomic features into consideration. We also defined univariate gene signatures that characterize the regulation of genes for individual genomic aberrations without considering inter-feature dependencies. A similar procedure was applied except that in the model only a single genomic aberration was used as the independent variable (Eq 2).

$$Y_i = \beta_{i,0} + \beta_{i,j} X_j \qquad \text{Eq2}$$

## Calculation of tumor-intrinsic genomic aberration scores

For each genomic aberration, we calculated a score that describes its pathway activity from gene expression data using the univariate and multivariate weighted gene signatures described in the previous section. To do so, we applied a rank-based statistical method called Binding Association with Sorted Expression (BASE) [68], which examines the expression of signature genes in each tumor sample to calculate a sample-specific score. First, we use median normalization on the gene expression profile. We then rank the expression profile in order of decreasing expression, $\mathbf{g} = \{g_1, g_2, \ldots g_n\}$, where n is the total number of genes. Then, we calculate a foreground f(i) and background function b(i) as shown below.

$$f(i) = \frac{\sum_{k=1}^{i} g_k w_k}{\sum_{k=1}^{n} g_k w_k}, 1 \le i \le n$$

$$b(i) = \frac{\sum_{k=1}^{i} g_k (1 - w_k)}{\sum_{k=1}^{n} g_k (1 - w_k)}, 1 \le i \le n$$

The foreground function captures the distribution of the highly informative genes, whereas the background function captures random distribution. The maximum deviation between these functions is then calculated. For the $w^+$ weights, the score is defined Score$^+$, and for the $w^-$ weights the score is defined Score$^-$. We then normalized the Score$^+$ and Score$^-$ scores by dividing them by their null distribution. The null distribution is calculated based on recalculating the Score$^+$ and Score$^-$ scores, but using a random ordering of the gene expression profile $\mathbf{g}$. This process is permuted 1000 times to generate the null distribution. The sample specific score for each genomic aberration is finally calculated as the difference between the normalized Score$^+$ and Score$^-$. More details on signature score calculation has been described previously [28,69]. Each patient in the Curtis and Ur-Rehman datasets was given a score for each of the 78 genomic aberrations, using both the univariate and the multivariate weights. A higher score correlates with a more deficient pathway due to a greater propensity for occurrences of the genomic aberration.

The scores based on the multivariate weights are used for our Random Forest or Cox proportional hazards models because they may benefit from the inter-dependent adjustments in the multivariate gene signature. On the other hand, scores defined by the univariate weights are used to show that our scores recapitulate driver genomic aberrations by themselves, as these scores more closely correlate with the aberration status when scores for the other genomic aberrations are not considered.

## Calculation of tumor-extrinsic immune infiltration scores

To calculate the six immune cell infiltration scores given the expression profile of a patient, we utilized a method described previously [70], which includes first defining immune cell-specific

reference gene expression profiles and then examining immune-specific gene expression in the patient gene expression profile. Essentially, a high score indicates higher infiltration of the corresponding immune cell type in the tumor sample, while a low score indicates low infiltration level. In this study, we focused on six major immune cell types that have previously been reported to have potential prognostic value, including naïve B (NavB), memory B (MemB), CD4+ T, CD8+ T, NK cells and monocytes.

## Calculation of Oncotype DX classes and scores

We used the genefu R package to calculate the Oncotype DX risk category (Onco-class) and score for a given breast cancer gene expression dataset [71]. Specifically, the function "oncotypedx" was used. The Oncotype DX score (denoted as Onco-score in the figures) is a number from 0 to 100, where higher scores correlate with a higher risk for distal metastasis. The Onco-class is directly calculated from the Onco-score (0–17 is low risk, 18–30 is intermediate risk, and 31–100 is high risk).

## Construction of random forest models to classify ER+ breast cancers into good versus poor prognostic groups

In both the Curtis and Ur-Rehman datasets, the data for prognosis was provided as time-to-event. For the Curtis dataset, the event was defined as disease-specific death. The Ur-Rehman dataset contained data on the time to recurrence and to distal metastasis. To maintain consistency between the datasets, we used "distal metastasis of disease" as the definition of an event for the Ur-Rehman dataset, since distal metastasis is a better indicator of poor prognosis and likely death due to disease.

To convert the time data to a classification problem, we defined a patient as having good prognosis when an event did not occur within 10 years of follow-up. Poor prognosis is defined as the incidence of an event within 10 years. Patients censored before 10 years are not included in the analysis. In the Curtis dataset, we counted 387 samples as having good prognosis and 293 samples as having poor prognosis. In the Ur-Rehman dataset, 168 samples were counted as having good prognosis and 177 samples were considered to have poor prognosis. We next built a Random Forest model using the R package "randomForest" to classify good vs poor prognosis with various combinations of features. We used a 10-fold cross-validation method, where the data is divided in tenths, and one tenth is used as validation while the other nine-tenths are used to train the Random Forest model. The tenths are cycled such that each tenth serves as the validation set once, and the area under the curve (AUC) scores from all the validations are pooled to generate the aggregate AUC score.

We also trained the optimized random forest model in one dataset and used that predefined model to predict prognosis in external validation datasets. The "predict" function in the "randomForest" library was used to extract the probability of a good prognosis for each patient in the validation datasets. These probabilities are then used to generate the receiver operating characteristic (ROC) plot and calculate the AUC scores using the R package "ROCR."

## Construction of Cox regression models for predicting patient survival and recurrence risk

To determine the performance of individual signatures and immune scores, we fit a univariate cox proportional hazards model for all 78 scores using the "coxph" function from the "survival" package in R and extracted the p-value and hazard ratios. To adjust for clinical variables, we included age, stage, lymph node status, grade, and size as covariates in addition to the

signature or immune cell scores in a multivariate cox proportional hazards model. To plot Kaplan-Meier (KM) survival curves, we used the median score as the cutoff to dichotomize low-score and high-score patients and utilized the R function "survfit" to plot the curve. The log-rank test is applied to calculate the p-value for the null hypothesis that no difference in survival exists between the two patient groups.

We also calculated the C-index (CI) to quantify the performance of our integrative, optimized Cox proportional hazards model compared to Cox proportional hazards models with just clinical variables or Onco+Clin as predictive features. The CI is calculated as the proportion of all comparable patient pairs with predicted risks concordant with their survival time (for example, a patient pair is concordant if the patient with higher risk has a shorter survival time). Note that patients are only comparable if both patients experience the event, or if one patient experiences the event before the other patient is censored.

To calculate the recurrence risk of patients in a validation dataset with a Cox proportional hazards model fitted in the training dataset, we used the "predict" function from the "survival" package in R to extract the predicted patient risks in the validation dataset. We then ranked the patients by their risk score and separated patients into three roughly equally sized groups to define the low, intermediate, and high risk groups. The KM method and log-rank test is again used to determine the performance of the model in defining risk groups.

### Optimization of classification and regression models

To determine the optimal combination of predictive features for the RF model, we performed backward selection by successive removal of features with the lowest importance as determined using the RF relative importance function. The AUC score is calculated after each addition/removal by a 10-fold cross-validation using both Curtis discovery and validation datasets. We then picked the predictive features with the highest AUC and the least number of features as our optimized model.

To similarly optimize the Cox proportional hazards model, we used the R function "My.stepwise.coxph" in the R package "My.stepwise" on the Curtis discovery dataset, with and without clinical features. This function performs a stepwise feature selection using the coxph model. The features chosen in the optimized model are shown in Table 1 as "Cox optimized model."

### Supporting information

**S1 Fig. Gene signatures do not generalize to ER negative breast cancer patients.** ROC curves showing that TP53_mut (**a**) and ERBB2_amp (**b**) predict driver aberration status at relatively low AUC scores in the ER negative breast cancer patients as compared to the ER positive patients.
(TIF)

**S2 Fig. Oncotype DX risk classes dichotomized by optimized model without clinical features.** Our optimized model's risk prediction without including clinical features is able to significantly stratify the Onco high (**a**) and intermediate (**b**) risk classes, and moderately stratifies the Onco low (**c**) risk class.
(TIF)

**S1 Table. Many individual gene signature scores are predictive of prognosis.** The p value was determined based on the hypothesis that the patients with good prognosis have different gene signature scores than patients with poor prognosis. The AUC was calculated based on the gene signature's ability to predict prognosis. The prognostic value shows whether higher

signature scores correlate with better or poorer prognosis. Onco-score is also included to compare with our signatures.
(DOC)

**S2 Table. Immune infiltration scores are predictive of prognosis.** Higher infiltrations of immune cells significantly correlated with prognosis predicts good prognosis, as expected.
(DOC)

**S3 Table. Relative importance of features in the Sig+Imm+Clin integrative model.**
(DOC)

**S4 Table. Signature and immune infiltration scores are predictive of prognostic risk.**
(DOC)

## Author Contributions

**Conceptualization:** Chao Cheng.

**Data curation:** Kevin Yao, Chao Cheng.

**Formal analysis:** Kevin Yao, Chao Cheng.

**Funding acquisition:** Chao Cheng.

**Investigation:** Kevin Yao, Chao Cheng.

**Methodology:** Kevin Yao, Chao Cheng.

**Project administration:** Chao Cheng.

**Resources:** Chao Cheng.

**Software:** Chao Cheng.

**Supervision:** Chao Cheng.

**Validation:** Kevin Yao, Chao Cheng.

**Visualization:** Kevin Yao, Chao Cheng.

**Writing – original draft:** Kevin Yao, Chao Cheng.

**Writing – review & editing:** Kevin Yao, Evelien Schaafsma, Baoyi Zhang, Chao Cheng.

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
