## [Decision Letter · Decision Letter 0]

15 Jan 2022

Dear Associate Professor Cheng,

Thank you very much for submitting your manuscript "Tumor cell intrinsic and extrinsic features predicts prognosis in estrogen receptor positive breast cancer" for consideration at PLOS Computational Biology.

As with all papers reviewed by the journal, your manuscript was reviewed by members of the editorial board and by several independent reviewers. In light of the reviews (below this email), we would like to invite the resubmission of a significantly-revised version that takes into account the reviewers' comments.

We cannot make any decision about publication until we have seen the revised manuscript and your response to the reviewers' comments. Your revised manuscript is also likely to be sent to reviewers for further evaluation.

Sincerely,

Feng Fu, Ph.D.

Associate Editor

PLOS Computational Biology

Sushmita Roy

Deputy Editor

PLOS Computational Biology

Reviewer's Responses to Questions

**Comments to the Authors:**

Reviewer #1: Yao et al described their work to address the heterogeneity in the prognosis in ER+ breast cancer using both genomic mutations within the tumor cells, and also extrinsic features in the tumor microenvironment. The study was based on TCGA and multiple datasets from METABRIC and GEO, and identified 72 genomic aberrations and 6 immunological features for the model development. The model was able to reach better performance than the traditional Oncotype DX assay. Overall, the study was well performed.

Following are some minor comments to be consideration.

1. As mentioned by the authors, adding imm features does not seem to improve the prediction model (0.817 vs 0.811, Figure 3e). Can authors comment on the additional benefit of including immunological features?

2. In the abstract, the authors should report the actual accuracy of different models for comparison.

3. The authors indicated that accuracy of the optimized model was much higher than the Oco+Clin model and the Clin model. It would be useful to report if this difference is significant or not.

4. In page 13, “In particular, patients in the high risk category had a hazard 3.196 times that of the intermediate risk category (p = 3e-11), and patients in the intermediate risk category had a hazard 2.523 times that of the low risk category (p = 0.001)”. It would be useful to use one group as the referent, and report the relative risk in the other two groups.

5. Please provide some brief description of the Curtis dataset and Ur-Rehman dataset in the Results section because the detailed information of both datasets were provided in later sections.

6. Please provide more informative description of feature names in the main text and Figures. For example, MemB is for Memory B gene expression profile.

7. Please define CI, ROC and AUC at the first place to use

Reviewer #2: In this study, the authors suggested that the prognosis of ER+ breast cancer patients can be inferred by both tumor-intrinsic (genomic aberrations) and extrinsic (immunological) features. They proposed a systematic framework that utilizes gene expression data for feature extraction and predicts prognosis of ER+ breast cancer patients. Based on TCGA data, 72 genes for tumor-intrinsic score, and 6 immune cell types for infiltration score were selected. Compared to Oncotype Dx, the most widely validated method for predicting prognosis for ER+ breast cancer patients, their framework showed the improved prognostic performance on external validation dataset. Further, they trimmed features to optimize their models and determined important features for predicting prognosis. This paper analyzed the prediction results extensively.

1. More detailed explanation about how to calculate a sample-specific score based on gene signature, and the extrinsic score based on immunological profiles should be added to the Method section, although the authors used methods developed by others. It seems that too many steps were omitted, compared to the Definition of weighted gene signatures for genomic events section.

2. In the Integrative models for classifying good and poor prognosis patient groups section, in Fig. 3f, and Supplementary Table 3, there were no clinical variables while the authors mentioned that the Sig + Imm + Clin RF model were used to determine which variables were significant. Make it clear.

3. In the Integrative models for predicting prognostic risk based on Cox regression section, specify a dataset used for analyzing 1,494 (747 High + 747 Low) patients in Fig 5.

Minor:

1. There are some typos. The followings are examples. The authors need to check the manuscript carefully.

Page 7: [39]Results -> Results

Page 17: An additional seven datasets => Additional seven datasets

Page 32: distinguishes -> distinguish

Page 34: In Figure 5, (d) is TNFRSF17_amp and (e) is MemB while MemB (d), and TNFRSF17_amp (e) in the legend. Make it clear.

**Have the authors made all data and (if applicable) computational code underlying the findings in their manuscript fully available?**

Reviewer #1: Yes

Reviewer #2: None

PLOS authors have the option to publish the peer review history of their article (what does this mean?). If published, this will include your full peer review and any attached files.

Reviewer #1: No

Reviewer #2: No
---

## [Decision Letter · Decision Letter 1]

17 Feb 2022

Dear Associate Professor Cheng,

We are pleased to inform you that your manuscript 'Tumor cell intrinsic and extrinsic features predict prognosis in estrogen receptor positive breast cancer' has been provisionally accepted for publication in PLOS Computational Biology.

Best regards,

Feng Fu

Associate Editor

PLOS Computational Biology

Sushmita Roy

Deputy Editor

PLOS Computational Biology

Reviewer's Responses to Questions

**Comments to the Authors:**

Reviewer #1: All the comments have been addressed.

Reviewer #2: The authors addressed all the comments.

**Have the authors made all data and (if applicable) computational code underlying the findings in their manuscript fully available?**

Reviewer #1: Yes

Reviewer #2: None

PLOS authors have the option to publish the peer review history of their article (what does this mean?). If published, this will include your full peer review and any attached files.

Reviewer #1: No

Reviewer #2: No

---

## [Editor Report · Acceptance letter]

4 Mar 2022

PCOMPBIOL-D-21-01740R1 

Tumor cell intrinsic and extrinsic features predict prognosis in estrogen receptor positive breast cancer

Dear Dr Cheng,

I am pleased to inform you that your manuscript has been formally accepted for publication in PLOS Computational Biology. Your manuscript is now with our production department and you will be notified of the publication date in due course.

With kind regards,

Orsolya Voros
